### A Global Model of Predicted Peregrine Falcon (*Falco peregrinus*) Distribution with Open Source GIS Code and 104 Open Access Layers for use by the global public

Sumithra Sriram<sup>1</sup>, Falk Huettmann<sup>2</sup>

<sup>1</sup>College of Computing, Georgia Institute of Technology, Atlanta Georgia 30332, USA <sup>2</sup>EWHALE lab, Inst. of Arctic Biology, Biology & Wildlife Dept., University of Alaska Fairbanks (UAF), Fairbanks Alaska 99775, USA

Correspondence to: Sumithra Sriram (sumithrasriram@gatech.edu)

Abstract. Peregrine falcons (*Falco peregrinus*) are among the fastest members of the animal kingdom, and they are probably the most widely distributed raptors in the world; their migrations and habitats range from the tundra, mountains and some deserts to the tropics, coastal zones and urban habitats. Habitat loss, conversion, contamination, pesticides and other anthropogenic pressures are all known factors that have an adverse effect on these species. However, while peregrine falcons were removed from the list of endangered species due to rebounding populations linked with the DDT ban in many nations of the world, no accurate global distribution models have ever been developed for good conservation practice and in an open

access data framework.

Here we used the best-available open access peregrine falcon data from the Global Biodiversity Information Facility (GBIF.org) to obtain the first publicly available global distribution model for peregrine falcons. For that purpose, we compiled over a hundred high resolution global GIS layers (1km pixel size) that incorporated various variables such as biological, climatic, and socio-economic predictors allowing to analysis habitat relationships in a holistic fashion and to

- build a generalizable model. These value-added layers have also been made available by us for the global public, free of charge, for further use and consumption in any modeling effort wanted (https://scholarworks.alaska.edu/handle/11122/7151). We created data extraction explicit in space and time also with an open source python script tool as well as with ArcGIS (via the GUI) on a PC. The obtained data cube (global, 1km pixel, 104 GIS layers) was 'mined' with the Salford Predictive Modeler (SPM) software suite, which offers one of the best platforms for data mining, to build the prediction model for
- robust inference. We found that peregrine falcons are widely urbanized occurring in coastal areas and also associated with riparian zones. This is the first model ever obtained using 104 predictors on a 1km scale predicting the potential ecological niche of falcons around the world. While our model might show uncertainty for parts of Siberia, Russia, it has an assessed global accuracy of over 95% and hence provides the currently best possible public available global prediction model for peregrine falcons, based on all available empirical data. Overlaid with the national parks of the world we found that most
- peregrine hotspots are actually located outside of protected areas warranting more protection efforts while global change unfolds. Finally, a nationwide assessment of the presence points taken from GBIF allows for insight as to the many signatory

nations that are still in violation of the open access data sharing requirement set by the Convention of Biological Diversity (CBD) and the Budapest and Berlin Declaration.

#### **1** Introduction

- 5 Predictive modeling explicit in space and in time has been used in ecology to generate distribution patterns for thousands of species to help model their ecological niches (Guisan and Zimmermann, 2000; Barry and Elith, 2006, Drew et al. 2011). These niches rely on data used and available. They are defined by a set of biotic and abiotic factors, and various methods have been explored for inference and the identification of relevant predictors (Breiman, 2001; Hastie et al., 2001; Peterson, 2001). Understanding these defining elements and concepts is important for successful conservation and species management, especially now, with the rapidly changing climate (Walther et al., 2002), shifting biomes and a large multitude
- of human impacts (Halpern et al., 2008). Species distribution modeling, both temporal and spatial, plays an important role in monitoring, managing and conserving

species distribution modeling, both temporal and spatial, plays an important role in monitoring, managing and conserving species effectively (Cushman and Huettmann, 2010; Drew and Perera, 2010). It is a convenient and effective method of dataanalysis, where identification of known ecological niches of species provides an insight into the presence or suitability of

- presence in remote (un-sampled) areas. For example, the fundamental niche was modelled for three quintessential Arctic bird species, and distribution patterns spanning over 200 years (from 1900-2100) were designed to predict the suitability of survival and changing spatial concentration corresponding to the changes in climate (Booms et al., 2011). By now, there have been many such distribution models that have put forth the ecological niches for various species over the years (Drew et al. 2011). Generally, the models built deal with the estimation of the fundamental ecological niche of the species in
- question. This can be achieved using either a mechanistic approach or a correlative approach (Soberon and Peterson, 2005). The mechanistic approach involves the physical modeling of the direct response of individuals and their metabolism, etc. to parameters such as temperature and humidity, and then using GIS to identify regions of positive fitness. The correlative approach, followed in this study, deals with using various predictor variables and building a predictive model using various supervised machine learning algorithms. On a global level, this is rather powerful because 'global correlates' can be
- identified, pursued and tested further. These correlations are not biased but are very powerful for inference. Using many data are therefore essential.

The ecological niche has been used and classified in many ways (Soberon and Nakamura, 2009; Cushman and Huettmann, 2010) - the Grinnellian niche of a species is determined by its habitat and its behavioral adaptations; the Eltonian niche is classified according to the foraging activity of the species; the Hutchinsonian niche, which is the most generic form of niche,

takes into account the various diverse environmental conditions and resources the individual requires to survive (Bruno et al., 2003). The range of such biotic and abiotic conditions that define the requirements of survival of a particular species is the fundamental niche. The complete set of locations that fit the requirements of the fundamental niche is the potential niche,

# Searth System Discussion Science Signate Data

and the set of locations where the species is actually found is the realized niche. Most models deal with the prediction of potential niches. Such modeling, however, has not been so prevalent with species that are as wide spread as Peregrine falcons. Global analysis platforms and computational solutions do not exist yet to deal with such large data, worldwide and on a 1 km pixel size, all as open access and open source for fast analysis. While the constraints were always put on species data, here we try to promote the opportunities on the 104 GIS predictors as open access and open source code to actually

#### 1.1 Biology of peregrine falcons

operate such a data cube effectively, e.g. on a local PC.

Peregrine falcons, though known for their widespread habitat range and adaptability, also migrate long distances connecting the winter with their nesting areas. Some of these nesting areas have been in use for over hundreds of years, and probably

- longer (Newton, 1979). Peregrine falcons can be classified into 19 subspecies depending on their geographic locations (Cade and Digby, 1982). Table 1 illustrates these known subspecies with their corresponding regions of occurrence. Poaching and hunting of this species has been ongoing for millennia, e.g. in falconry; perhaps it was somewhat sustainable even. But in the mid-20<sup>th</sup> century, the peregrine falcons were critically endangered globally, and were even close to extinction in North America, due to the excessive use of DDT and other chemical pesticides that led to their death or
- reproductive failure due to the thinning of their egg shells (Newton et al., 2008). From the eventual ban on the use of organochlorine pesticides onwards, and with widespread reintroduction of these species and protection under various national and international legislations, they have since made a strong recovery in many parts of the world (Tordoff and Redig, 2001; Jacobsen et al., 2007). Known for their speed and broad geographical availability due to their adaptability, they are probably the most frequently used raptors for falconry. The detrimental effect of the pesticides aside, this aspect of
- human pursuit has made these falcons more vulnerable, being pursued and poached by egg collectors and falconry thieves alike with the ever-present demand in the Middle-east. Some of the international protection policies that have been established to ensure the protection of these species have been listed in Table 2. Apart from international regulations, individual countries have declared their own additional laws that protect these species from harm, a few of which are listed in Table 3.

#### 25 **1.2 Conservation efforts**

The Convention of Biological Diversity (CBD) plays a central role in modern times also using digital opportunities. It is therefore considered here in more detail. Specifically it deals with online data aspects of biodiversity conservation affecting world-wide conservation management. CBD is an important multilateral treaty signed for now by 196 parties from around the world for sustainable development. One of the key points still discussed at the 10<sup>th</sup> Conference of Parties (COP), held in

Japan, is the issue regarding sharing of data on biodiversity (Balmford, 2005). Often, the areas that are richest in biodiversity are also the ones that lack the resources for conservation, and enough data for analysis often is unavailable to make good decisions. Hence it is important for scientists handling databases, in public office and such funding, to make this data

# Searth System Discussion Science Signate Data

available for all users and researchers around the world to help build robust and collaborative methods of conservation that will ensure holistic benefits (Huettmann, 2011; Resendiz-Infante and Huettmann, 2015). In this study we also examine the empirical data that is available for peregrine falcons, when placed against the predictive models obtained, in order to find the countries that are in good compliance with this agreement of data sharing.

- By now, the peregrine falcon is one of the best known examples for 'synurbization', the adaptation of wild animals to the rapidly invasive urban conditions, since its reintroduction (Luniak, 2004). The increased number of urban pigeons presents a central role in this discussion as prey species and in populated settings. Modeling, predicting and studying the distribution pattern of such a global and adaptive species can give useful insights into ecological aspects such as the effects of globalization on biodiversity and wilderness habitats. So far, one will find several distribution maps put forth by various
- organizations, showing generic but often conflicting habitat regions for these birds, none of which really carry relevant and compliant metadata, scientific accuracy metrics, are not repeatable and are not available for a repeatable scientific assessment in a useable GIS format (Huettmann, 2004; Huettmann 2015b; Zuckerberg et al., 2011). The global distribution patterns of peregrine falcons have not been studied in detail for conservation purposes since its

removal from the U.S. list of endangered species in 1999 and the rebound due to DDT ban and breeding programs. Modern

- study methods have not been employed, yet. In this study, we investigate steps to achieve the first global distribution model for peregrine falcons, using over a hundred compiled open access predictors that include climatic, biological and socioeconomic factors to represent a more holistic set of factors that can have an effect on the survival and suitability of the species in the region. We consider the species – *Falco peregrinus* (Taxonomic Serial No.: 175604), which encompasses all the subspecies, to build a general global niche (whole year round). Further, here we try to present a software open source
- analysis platform done in python code for such analysis cases for generic uses of this data cube readily to be used by the global public for their own purposes. We believe this is a rather large progress because such 104 data layers do not exist yet in a readily available GIS format as provided here. It allows to demand for best-possible holistic views in any habitat study.

#### 2 Methods

#### 2.1 Training data (presence and pseudo-absence)

- We used the 'presence only' data for peregrine falcons from the Global Biodiversity Information Facility (GBIF.org). As per Convention of Biodiversity (CBD), it is the one-stop open access international data warehouse for species occurrence. Many nations confirmed their participation by signature and ministerial support. GBIF represents currently the largest known empirical data about peregrine falcons in the public realm, which includes information such as the geo-coordinates of the location, the date, and the organization that reported the record (e.g. a sighting or a specimen). This raw data had to be
- filtered for accurate and duplicate records, for records with incorrect geo-referencing and for records with ambiguous data to finally obtain 60,261 unique presence points, less than half the size of the raw database. Once the presence points were established, we then plotted 35,800 evenly distributed random points of pseudo-absence over land masses (excluding

Antarctica), using the "Create Random Points" tool in ArcGIS. This was done to obtain a representative pseudo-absence data layer for the world.

Figure 1. Global distribution of Peregrine falcons (presence points)

- 5 Another major aspect of this study was the first-time compilation and public delivery of over hundred global GIS layers at a 1 km x 1 km resolution from various open source projects for use as predictor variables for such models. We followed the initial work by Ohse et al. (2010) and Herrick et al. (2013). The range of predictor layers used is presented in detail in Appendix Table 1 and they include the following:
  - Climatic data such as mean monthly temperatures, mean monthly precipitation, mean monthly solar radiation, and global aridity index
- 10
- Bioclimatic variables (bio 1 bio 19) as defined by WorldClim
- Digital Elevation models (DEMs) and other variables derived from them such as slope and aspect
- Variables pertaining to biodiversity such as species richness of birds, mammals, amphibians and plants, annual average potential evapo-transpiration
- Quantitative indices indicating the effect of humans on biodiversity such as the Human Influence Index (HII), Human Footprint level and Last of the Wild
  - Proximity measures to coast, rivers and roads, which were calculated using the Euclidean Distance tool in ArcMap
  - Socio-economic factors such as Gross Domestic Product (GDP), human population density and count, Infant mortality, literacy rate, life expectancy, and trade and night light pollution.
- Variables such as density of livestock population such as pigs and poultry were also included, which prove to be highly influential factors when dealing with species that live in close contact with humans.

5

These variables were gathered from various open source projects, were re-projected and re-sampled for better alignment (e.g. for coastline and for each other) allowing us to deliver this value-added data product. The global layers are available for open access in GeoTiff format (LZW compressed), in WGS-84 projection, with a resolution of 1 km x 1 km in our public repository dSPACE UAF library and Google Drive (available upon request from the authors). This dataset has a size of 37.5 GB. They can also be easily converted to/from ESRI grid and ASC formats. Using such a wide range of predictors helps us

to start explore and recognize the hidden but so far unknown but driving factors that influence the species. In predictive models, having a complete description of the ecological niches is essential and reduces uncertainty whereas parsimony fails (Elith et al., 2006; Guthery et al., 2005).

Usually, the compiled layers were loaded into ArcMap and then overlaid with the presence and pseudo-absence points compiled. Using the Extract Multi-values to Points tool on ArcMap, the appropriate values for all these layers at the aforementioned points were extracted, and this compilation of data was used for creating the distribution model. But here we developed a second and open source approach and making it available to the global public for their empowerment and to use these data more effectively for their own purposes: The extraction of values from this data cube can also be done using python and its supporting libraries. Python is rapidly becoming one of the most popular languages used for machine learning

- and any advanced analysis (Harrington, 2012). The extensive libraries and packages available are programmed to do most of the 'heavy lifting' and provide efficient models and solutions, enabling users to concentrate on the problem at hand rather than the modeling specifics. It also gives users the powers to determine the predictors that are used to build predictive models (as per Leo Breiman 2001). The script, that is available for access for the global audience, can be used as a generic template to handle big data on small machines as well (IBM PCs here). This, when combined with other useful multi-
- processing libraries can be used to scale up performance when run on the cloud or clusters, as needed for in-time applications for instance.

#### 2.2 Modeling approach

We used primarily the TreeNet algorithm in SPM7 provided by Salford Systems Ltd (https://www.salford-systems.com/) to build the distribution model. We also tested RandomForest in comparison. These algorithms have been widely used for modeling by data mining of ecological data for conservation management (Craig and Huettmann, 2009). They are all known to generate highly accurate models for both regression and classification and are also pretty robust when dealing even with faulty data and outliers (Fernández-Delgado et al., 2014). SPM also gives the user the flexibility of controlling the parameters of the models. The classification models were trained to predict the relative index of occurrence (RIO) of peregrine falcons in any given region of the world using the presence and the pseudo-absence points with all attributes from

30 the data cube. We used the 'balance' class weight option to balance the unequal presence and pseudo-absence sample sizes and kept all others at 'default' (a setting known to perform very strong).

#### 2.3 Display of prediction surfaces

Next, using the Create Fishnet tool in ArcMap, we generated a global layer with an equally spaced point lattice grid with a 1 km x 1 km resolution, bounded by the continental landmass. This was then overlaid with the hundred layers of predictors, and the technique that was used to extract the values of these variables to the presence points was also used to achieve the same with these points in the fishnet layer. This set of points was then 'scored' using the classification predictor model built

5 same with these points in the fishnet layer. This set of points was then 'scored' using the classification predictor model built in order to obtain the global distribution of the relative index of occurrence of peregrine falcons. We then used the Inverse Distance Weighting (IDW) tool in ArcMap to create the raster surface for the predicted RIOs of presence.

These values of RIOs were then extracted for the known presence points that were used to build the model. A frequency distribution histogram was obtained for the RIOs to show the range of indices that predict the presence of the species in the region, according to the model built. The error percentage of the model is then used to determine the cut-off threshold of the

indices to obtain a binary presence/absence prediction for peregrine falcons.

#### 2.4 Accuracy assessment

The accuracy of the models obtained was assessed using the Relative Operating Characteristic (ROC) - Area under the curve (AUC) metric, as is commonly used (Pearce and Ferrier, 2009). The ROC consists of a graph of a binary classifier that plots

- 15 the true positive rate against the false positive rate. We assumed AUC scores less than to 0.7 indicate low accuracy, between 0.7 and 0.9 to indicate moderate accuracy, and scores higher than 0.9 for high accuracy (Swets, 1988). We also obtained another set of the few publicly available Open Access presence points from MoveBank (www.movebank.org) that we used to validate our models. Though there were over fifteen datasets that were listed for Peregrine Falcons, only two of them were available for public access, and none of them were shared with GBIF. Extracting the predicted RIOs at these points and
- 20 plotting their frequency distribution allowed us to examine the accuracy of our model for many regions. The workflow in its entirety is illustrated in a flowchart shown in Figure 2.