# Peer review of "A Global Model of Predicted Peregrine Falcon (*Falco peregrinus*) Distribution with Open Source GIS Code and 104 Open Access Layers for use by the global public"

_Earth System Science Data, 2016_

## Short Comment (SC1) · 18 Feb 2017

Hi, it is good to know about above hundred variables in a model. I wonder about the generality of this complicated model and the transfer-ability.

---

## Referee Comment (RC1) · Anonymous Referee #1 · 2 Apr 2017

Difficult data set, provocative presentation, but potential highly useful. Attempts to set a very good open access example. Uncertain how it fits for this journal, but more uncertain how it would it fit better anywhere else. With some changes and edits could be a very useful product. (Not easy to review without page numbers!)

From the text as provided - and following Figure 2 very closely - this reader understand four purposes:

1) Compile a clean presence / absence database for peregrine falcons; 2) Compile a diverse but consistent database of potential predictor factors; 3) Through iterative

scoring (weighting) and modelling, develop a predictive indicator of falcon occurrence; and 4) Validate the predictive model by comparison with independent falcon distribution data.

Thinking of subsequent research users, this work should give them enhance capabilities to:

a) repeat the same analysis for a different species, avian or terrestrial? b) from these data, test alternative hypotheses or outcomes for falcons? c) from this basic outline and recipe, try to update the case for falcons using more recent predictor fields?

Thinking of resource managers, this work should provide them some tools to:

a) better monitor falcon populations and distributions? b) develop better or alternate conservation strategies?

(I suppose these researchers and resource managers constitute the "global public" in the title?)

Whatever their purposes, these subsequent readers and users should be able to take advantage of and rely on:

a) the clean presence / absence data (perhaps in two versions, one without and one with the pseudo-absences); b) the compiled and georeferenced predictor layers (perhaps a users could select some subset of those layers); and c) a complete recipe with links to all resampling, extraction, thresholding, etc. software tools (contained in this manuscript?).

Authors provide link to very good ScholarWorks site at University of Alaska. There one finds all the predictor layer files individually, plus a reference table (same as Appendix A) and a Python code. But the presence / absence files from GBIF seem missing? And, although the url references a 'handle' identifier system, this reader understands 'handles' as allowing changes to metadata or data covered by that identifier? A reliable location but flexible content? Not sure how the ScholarWorks system fits the DOI

requirements of the ESSD journal? Authors or editors will know more about this than this reviewer? Should we have a separate more-permanent snapshot of the essential files, saved as a backup somewhere else?

How does a user get access to the presence / absence data? Back to GBIF for each use? But the authors describe (section 2.1) substantial reprocessing of GBIF data, to remove half of the raw data. A user needs access to that product.

The authors have provided a useful snapshot. But, as they describe in their text, the falcon populations, their relative health, and the global environment all evolve. The authors should perhaps make more emphasis of the transitional nature of these predictors. They make valid points about urbanisation, but based on quite old data and with little sense of what next? They do comment on conservation options, but mostly in retrospection? What new or different information would users get if the authors used more up-to-date source files. For example, economic and demographic data, mostly from a year 2000 publication (Newsweek / ESRI) which reports data from late 1990's. Climate data from well before 2005? Some data layers come from more recent publications, e.g. 2013. If, as I suspect, the authors have provided a remarkable product by assembling the best or most available - to them - data sources, they need to justify this in the text and to then comment about how they or others might update all or specific layers with fresher informations (e.g. new land use data sets from ESSD). This reader also wonders about the extensive incorporation of light, radiation and clouds in the predictor layers when only night-time light and September solar radiation turned out to have predictive value?

If interpolating any of these layers to 1 km by 1 km spatial resolution was easy, many of the sources would already have performed that downscaling. Here, authors describe a 'push-the-button' sort of downscaling technique, that takes all layers from original resolution to 1 km by 1 km. Apparently this happened through a 'create fishnet' tool in ArcMap? Users need more information about this downscaling step. For example, when this reader searched on '1 km' among the ESSD data sets, I found 1 km DEM for

[Figure]

Antarctica, 1 km satellite source data for lake and river water temperatures in Europe and 1 km projections for rainfall in UK. One could imagine and hope that some future researchers would want to compare the UK 1 km rainfall patterns with the 1 km predictor fields produced here and to then compare the falcon distributions data likewise. But have these authors given those future users enough information to understand these downscaled data?

To my list of four purposes above, the authors seem to have added a fifth:

5) Evaluate and discuss national compliance with CBD conventions and data contributions to GBIF. That discussion, although valid, does not add to this manuscript and perhaps belongs elsewhere?

Several small comments but too hard to submit those without page numbers as reference. Reviewer or editor can address those in a subsequent version.

---

## Short Comment (SC2) · 11 Apr 2017

This paper is completely out of touch with peregrine biology and the ecological drivers behind peregrine distribution. Figure 4 indicates that peregrines are associated with areas with low human infant mortality, high population density, and lots of light pollution. No explanations are given for why there might be more peregrines in areas with low infant morality and higher population density, although I suspect it has more to do with the fact countries with lower infant mortality rates are affluent. These are the countries that banned DDT and have programs that look for peregrines. Does anyone think

that peregrine falcons actually prefer light pollution? If not, then we should not use light pollution to predict where they may occur or what places need to be protected. Without an understanding of the biological mechanisms that underlie the models and what sources of confounding might be present, the inferences are useless at best and misleading at worst.

The main conclusion of the paper, that few peregrine falcons are present in protected areas and, therefore, we must protect more areas is also flawed. The recovery of peregrine populations is a conservation success story. Based upon their own models, peregrine falcons are predicted to occur virtually EVERYWHERE outside of Siberia, the Gobi Desert, the Sahara Desert, and the Greenland ice sheet. If we believed their model had real predictive value, we would prioritize the protection of areas with more people and more light pollution and we would search out areas of low infant mortality. In reality, populations of peregrine falcons are recovering because the relevant protections (namely the banning of DDT in developed countries) are already in place.

Unfortunately, important ecological drivers, such as the presence of nesting structures (mostly cliffs) or the use of DDT are not included. In short, there is nothing in this paper that is useful for someone that actually studies or needs to manage populations of peregrine falcons.

John Citta, PhD, Alaska Department of Fish and Game, 1300 College Road, Fairbanks, AK 99708

---

## Short Comment (SC3) · 20 Apr 2017

While the idea to develop a global map of peregrine falcon distribution is a worthy goal, in my opinion, this paper has several major problems that make the results unreliable without major revisions.

**The source data are non-representative**

The data used in the analysis were collected from a variety of sources in a non-random way that leads to a dataset that is not representative of the entire distribution of peregrine falcons worldwide. Peregrines were more likely to be observed and reported in certain countries and certain locations. Because no effort appears to be made to account for the potential biases these data can introduce, this paper does not actually model the distribution of peregrine falcons. Instead, this paper models the distribution of peregrine *sightings that have been reported to GBIF*, which is a potentially very different distribution from the actual global distribution of peregrines.

By comparing data collected non-randomly from certain countries with random locations spread across the globe, the authors created a situation in which country-specific socioeconomic factors can erroneously be given explanatory power, whereas environmental variables that biologists know to be important in affecting peregrine distribution are not included. We know peregrine falcons do not select areas based on human infant mortality rates. Although it is possible that there is a strong unexplained spurious correlation with high predictive power between the two variables, a more plausible explanation is that peregrine falcons are more likely to be observed and reported in countries with low infant mortality rates. Therefore, it seems likely that the country-specific socioeconomic factors dominating the model actually are related to levels of reported sighting rates rather than to global occurrence. Many of the areas they determine have low probability of occurrence are remote areas with low human populations and, hence, little opportunity for online reporting. The authors discuss the fact that reporting varied dramatically among countries but do not explain how they dealt with this issue. The countries that they identify as lacking in data ("Russia, China, Brazil, and some African nations") correspond well with the few areas they identify as lacking peregrines (Figure 7), even though there is ample evidence that peregrines nest widely across Russia and into China. This disparity provides more evidence that these country-specific differences in reporting were erroneously carried over into their distribution map.

**The resulting model does not appear to fit the testing data well**

Because the authors use a data-mining technique with 104 variables, they are able to adequately fit a model to the training data. However, because they do not deal with underlying limitations of the data or use appropriate biological variables known to be important to peregrines, the resulting model performs poorly with the testing data (Figure 9). Although the authors claim a 98% accuracy rate, Figure 9 really shows a poor fit to the predicted model (a good fit would look like Figure 6). Although availability is not shown on this histogram to assess the fit properly, the data appear to perform little better than random points (i.e., if 2% of the land mass is predicted to be unoccupied, then 2% of random points would be expected to fall in that area by chance). Even with the very low threshold for a location's being considered occupied (>0.01 predicted index), the testing data still cross several areas not expected to have peregrines based on their map (Figure 8). In addition, the measure of model fit needs to be

modified to take availability into account (e.g., if 98% of the area is identified as having peregrines, even random points would be expected to have a 98% accuracy rate using their metric).

**The predicted map in Figure 3 does not appear to correspond well with some other smaller-scale peregrine distribution maps**

Figure 3 should be compared with existing maps of peregrine falcon distribution, where available, to gain some sense of how well the models actually work. Distribution maps synthesize expert opinion and multiple data sources to summarize local distributions. If this new map differs greatly from other maps based on a variety of sources, that difference should be explained by the authors. Table 5 attempts to compare the map to a previous description of peregrine distribution in Russia, but it does not give enough information to compare the modeled distribution adequately with the previously described distribution. The first comparison is too general and deals with population density, not occupancy, the second comparison was done at a spatial resolution different from that of the model, and the third one does not support the modeling results. Peregrine falcons do migrate across much of the area in Russia identified as not used (Dixon, A., A. Sokolov, and V. Sokolov. 2012. The subspecies and migration of breeding Peregrines in northern Eurasia. FALCO 39), again suggesting that the model does not fit independent data.

**Combining different types of data**

Although the composition of the testing data is not adequately described, these data appear to include breeding, non-breeding, and migratory locations. How can those types of data be fit with the same model when birds are selecting for very different conditions at different times of their annual life-cycles? A bird flying over the landscape during migration is selecting for very different variables than a bird selecting a nesting location; consequently, lumping these life-history stages in analyses is unlikely to uncover useful patterns and will likely result in incorrect results. To be accurate and useful, three different maps (breeding, non-breeding, and migration) should be developed, analyzed, and combined.

**Other comments/questions**

- The justification for using the 0.01 cutpoint for determining occupied pixels is not adequately explained. This cutpoint seems to be excessively low and, therefore, would be expected to erroneously include many areas that actually are not used.

- Because the authors claim that peregrines select areas with large populations and high light pollution, it is not surprising that national parks do not contain the modeled habitat. However, we do know that many national parks contain peregrine falcon populations and that peregrines use areas without light pollution over much of their range. This latter point is especially true at high northern latitudes.

- Check Figure 10, a visual comparison Figure 7 to a map WDPA protected areas suggests that many protected areas do appear to be in areas with predicted occupancy by peregrines.

- Give units for pixel size column in Appendix A.

- Although the authors say they use a 1 km x 1 km grid for all of their variables, the text should make clear that many variables are country-specific variables.

- How can the same model be fit to mean October and November temperatures in both the Northern and Southern hemispheres?

- Antarctica was removed from the analysis; similarly, the Greenland ice sheet is not potential habitat and also should be removed from the analysis.

- The methods say that 60,261 presence points were used, but Table 6 says there were 578,256 presence points in GBIF just for 11 selected countries. What explains this discrepancy? Please explain specifically which data were selected and how they were screened.

**Conclusions**

Data-mining has many useful applications, but it still requires unbiased data (or existing biases have to be adequately accounted for), the explanatory variables chosen have to explain the underlying processes and be relevant to them, and the resulting model needs to show good agreement with independent data (testing data and previous distribution maps). As currently written, this paper does not meet those three criteria.

---

## Author Comment (AC1) · 8 Jun 2017

1. Generality of the model and the transferability

Our reply: Thanks, this comment came from the public and it needs to be said that our model is already global, thus it generalizes worldwide (more is not possible, than a global inference)! The model presented here is a generic one that generalizes across a compilation of diverse predictor layers. All available global peregrine falcon data were used as an example to illustrate the use of the unique data cube compiled, and the case

study assessments clearly show for everybody to see that the model obtained is not only accurate, but also highlights new, important and so-called unconventional factors not previously known influencing the distribution of these species in the Anthropocene. This is new scientific information and a template on a global scale, to be extended to any other terrestrial species for obtaining a general and unbiased distribution model for better conservation management. We see no relevant bias or lack of generalization in that. Some of such work was already started by us earlier, too (e.g. Huettmann et al. 2011, Kandel et al. 2015). Thus, our model is already fully generalizable, and applied, worldwide, to any pixel in the world! We feel, the author of this comment did not really understand that concept of global model predictions and such an inference (as we argued above).We showed in the maps assessments about how well those models perform and generalize (98 % AUC), globally. This is virtually unachieved other than this study. So we think we have addressed this comment in our work to good satisfaction.

2. The source data are non-representative

Our reply: Thanks, but that cannot be so correct because we use best-available data world-wide; that is true for the 100 (!) GIS habitat predictors (all describe the habitat pixels where falcons occur), and for the 60,000 falcon observations worldwide (these are sites where the falcons where actually seen, presence; unbiased). These are the best data at hand for such work and this species ever used! So we have used all the available data that were available and see no bias. We discuss all relevant details of these topics in our manuscript. We think this comment tries perhaps to speak to the fact of survey effort; and normal distribution, parametric assumptions. We have clearly shown and stated in the MS text that these are smaller issues in our work, for instance due to the high AUC assessments and non-parametric data mining algorithms that can account for parametric assumptions. To further help this reviewer and comment, we suggest to read work done by Kadmon et al. (2004) regarding so-called road biases and why this is no problem in the methods and approaches we use. The use of proxy

layers is widely used, and indeed, we use such predictors to great success. This is well known and applied for decades now. So we see no issue on non-representativeness here and we feel that our text speaks to those facts fine.

3. The authors created a situation in which country-specific socioeconomic factors can erroneously be given explanatory power, whereas environmental variables that biologists know to be important in affecting peregrine distribution are not included.

Our reply: Thanks, but peregrine falcons are migratory and cross many nations. This is not caused or created by us. We used 100 globally consistent GIS layers designed for global analysis, but we used the actual pixels for our analysis scale. So we do not have a national bias or artefact; nobody has ever used a global pixel-based analysisfor peregrines. So to argue we have ignored predictors, or we would run a national bias, sounds very odd to us, and when a global analysis is the aim. It's just not true. Our GIS layers act as direct, and proxy predictors along the flyways and ranges of this species. We wish to add more, if we are told which ones that would be (the reviewer left it as an unsubstantiated statement). The models simulated were un-biased, impartial and run without any intentions other than to obtain best possible predictions for inference (that's Leo Breiman 2001, basis of machine learning). All 104 layers, including commonly used environmental variables, were included in the analysis. Repeated runs of the models consistently picked the variables mentioned in Table 1 to be the most influential factors that determine the predicted distribution of these falcons. We speculate that this comment might be driven by the idea that falcons would just occur in the 'wild', and are not urbanized. Our models, and all recent literature shows entirely the opposite for a global analysis and for this species. This is easy to show, as provided in the MS text, e.g. release of hand-raised habituated individuals while the wild species DNA/stock became extinct and urbanization on the rise. We still have (sub) samples from virtually all areas of the globe in our assessment; our point sample map shows that clearly!

4. The resulting model does not appear to fit the testing data well – cutoff point of 0.01

Our reply: Thanks, we think the reviewer does not really understand that we use 'classification trees', recursive partitioning, but not logistic function 0 to 1 which is symmetrical. Trees are none of that! It's the fundamental difference from frequency statistics; our methods are widely published, e.g. see Kamel et al. 2015 and Mi et al. 2017 etc. The cut-off is non-symmetric, rightly so, and simply stems from the assessment with alternative testing data. This is done in machine learning for the best predictions, and the best inference. Whereas the 'model fit' (like an r2) is 100% not relevant in that, as per Leo Breiman 2001 (inference from predictions, but not from model fit requirements). The ROC curve of the model chosen shows an accuracy of 98 (2% error, globally!)%. That means the data are to 98% reproducible, worldwide; it's a world record and certainly for this species! The RIOs of all the present points were extracted, and to accommodate the 98% accuracy error rate when compared with real data! Thus, 2% of the points on the lower side of the RIOs were considered to be erroneous (easily >95% of the alternative data correctly classified; beyond significant!) and this results into a threshold of 0.01. Such high accuracy hints at an almost perfect recreation of the training data set, and to be sure and spatially precise, other validation methods were performed too for a better proof of evidence. The testing data was obtained from an independent source, and when the RIOs w.r.t the model were extracted, it was observed that less than 2% of the points had a RIO of less than the threshold value, which again, adheres to the 2% error assumption. It's an alternative test matching from what we initially stated. These methods are widely done and published, e.g. Elith et al. 2006, Huettmann et al. 2007, Kandel et al. 2015.

5. The predicted map in Figure 3 does not appear to correspond well with some other smaller-scale peregrine distribution maps.

Our reply: These matches exactly because they are based on each other. Figure 3 is the base raw heat-map obtained from the model and shows the relative index of occurrence (RIO; no threshold). The actual distribution map interpreted from this model for peregrine falcons is shown in Figure 7 and it is EXACTLY based on Figure

3 and its structure in the legend and pixels. So we kindly disagree with this comment by the reviewer. We are happy to send over both maps for the check; as needed. Here we offer in our paper the raw model as well as the interpretation, and then we show how well those work by using alternative data (as outline in previous reply points and citations we provided for the evidence).

6. Combining different types of data -breeding, non-breeding, and migratory locations

Our reply: Right, this is exactly what was done. The paper mentions that the model provided for these falcons is general and global, and includes all the areas that are suitable for these falcons year-round. We refer to the GLOBAL ECOLOGICAL (YEAR-ROUND) NICHE. That's exactly our aim and achieved here with a very high accuracy. We fully agree that a nesting niche, or a wintering niche is located within the text should be very clear on this.

7. Give units for pixel size column in Appendix A.

Our reply: Thanks, we changed it in the manuscript. Thank you for pointing out the error.

Changes in the manuscript: Updated Appendix 1, "Original pixel size" column, with units.

8. How can the same model be fit to mean October and November temperatures in both the Northern and Southern hemispheres?

Our reply: Thanks, we run a global model, so we use global predictors for every 1km pixel on earth. Sure there are different processes and things going on by hemisphere and season etc. However, our 104 predictors catch that fine, certainly the global climate model predictors. We do infer from the predictions, explicit in space and time. Thus, what the reviewer describes is 100% not a problem (this would be a potential problem when inference is done from parametric assumptions and model fits; but that's exactly NOT what we do). Our predictors describe the pixels and their climate explicit in space

and time (see Worldclime for an example, or any other global model; let's say Wei et al. 2011)

9. Explain specifically which data were selected and how they were screened.

Our reply: The raw collection of data points included almost 500,000 points, of which duplicate records, records with incorrect geo-referencing and records with ambiguous data were removed. Also, only points that were reported after the year 1990 were taken into consideration. Our process results into a conservative set of valid peregrine sightings. We are happy to send the reviewer the raw data and our test set. All of this is fully transparently done based on the GBIF data set.

Changes to the manuscript: (Section 2.1) "This raw data had to be filtered for accurate and duplicate records, for records with incorrect geo-referencing and for records with ambiguous data to finally obtain 60,261 unique presence points, less than half the size of the initial raw database."

10. Check Figure 10, a visual comparison Figure 7 to a map WDPA protected areas suggests that many protected areas do appear to be in areas with predicted occupancy by peregrines.

Our reply: Thanks, we can let others tosee and decide well. Please have a look: It's pretty clear that the protected areas are not where the prediction hotpots are, and our geo-referenced overlays (done in a GIS thus reliable) show just that, without relevant margins of errors. The trends are pretty clear. Due to the "synurbanization" of the falcons (which is widely published, citations provided, and explanations given earlier above), the falcons are flocking more towards urbanized areas. This has been captured correctly by the model, and by the reported points in GBIF. Conservation measures for protecting these birds need to be changed accordingly to accommodate this change. A classic example is found in Moscow/Russia, for instance, where the Stalin architecture buildings became a cliff, host for urban peregrine nests, and those birds there are active at nights and feeding on pigeons! Our model shows exactly such things!

11. Although the authors say they use a 1 km x 1 km grid for all of their variables, the text should make clear that many variables are country-specific variables.

Our reply: Thanks, sure, this has been added to the manuscript to be clear. Certain layers like the socio-economic layers are available only at a country and county level resolution for privacy concerns of social scientists. Such layers have simply been sub-sampled for alignment with the rest of the layers in the model and its pixels. So we find we have addressed all of this is fine in the MS text, as is.

Changes in the manuscript: (Section 2.1) "Certain layers such as the socio-economic layers were available only at a very coarse resolution, due to privacy concerns. Such layers have simply been sub-sampled for alignment with the rest of the layers in the model and all its pixels."

12. Antarctica was removed from the analysis; similarly, the Greenland ice sheet is not potential habitat and also should be removed from the analysis

Our reply: Thanks, our model correctly predicts that the Greenland ice sheet is not potential habitat or part of the ecological niche (while parts of coastal Greenland are; that is well reflected in the species literature). We find, the continent of Antarctica, ice-covered, can be excluded (but we kept the southern Antarctic islands and regions) . So we think this critique is addressed in our work.

Lastly, we think that another key aspect of this manuscript, the delivery of 104 (!) GIS layers for the public all done Open Access free of charge has not received sufficient appreciation by those comments. So we wish to emphasize that further, and beyond 'just' the falcon model (which is great by itself and with such a great accuracy, all updated as stated!)

---

## Editor Comment (EC1) · D. J. Carlson (Editor) · 19 Jul 2017

The manuscript ESSD-2016-65, addressing global distribution of peregrine falcons, presents one of the most difficult decisions this journal has made.

The authors have assembled data from many sources. They have made all their sources explicit and all the data, databases, tools and algorithms reliably and openly accessible. In this openness they set a high standard that resonates with the ideals of this journal. They push the boundaries of ecology and indeed of earth system science.

[Figure]

Their work crosses a wide range of disciplines and thereby challenges the evaluation processes of this journal to provide competent reviewers and reviews across that range of topics.

Despite this breadth, this work has unfortunately failed to assure potential users of its fundamental quality. In particular it has not adequately addressed two concerns: that the distribution data themselves reflect a substantial observational bias and that the environmental data, particularly the meteorological data, do not support utilisation at 1 km resolution.

Reviewer 1 raised the resolution issue. The authors have not responded to this issue. As this editor knows, and as many papers in this journal demonstrate, the interpolation of any global environmental data to 1 km resolution remains far beyond the reach of the research community. In this journal one can find 1 km data for rainfall over the UK or very high resolution aerosol data over Germany, but no global variable of any type or source at 1 km. ECMWF, a leading global centre for atmospheric reanalysis, confronts severe validation challenges as it prepares global reanalyses at 11 km resolution. The authors fail to provide any rationale or validation for their interpolations. By this failure they invite, unfortunately, disdain for or dismissal of their work.

The issue of bias in detectability arose from two comments submitted during the open discussion. Although the authors submitted a reply, this editor feels that their reply took a somewhat orthogonal direction from the original concerns. One of the researchers who submitted an open comment has responded to my request for further comment as follows:

"Thank you for allowing me to respond to the author's comments. In my opinion, this is not a question of data-mining versus avian ecology, but an inappropriate application of data mining. My main criticism of the paper is that the peregrine locations are biased by country-specific differences in reporting rates. This bias in the location data results in country-specific socioeconomic variables being incorrectly influential in the model and

therefore, the resulting prediction map is also suspect. I do not believe the author's response adequately addressed this criticism.

The authors state "that cannot be so correct because we use best-available data worldwide" and "these are sites where the falcons where actually seen, presence; unbiased". Clearly, these assertions are not adequate proof that the locations are unbiased. Data can be the best-available and represent actual locations and still be biased by factors such as differential reporting in certain countries and areas. In fact, the manuscript states that certain countries had low reporting rates and a comparison of their map to other data sources indicate that these countries with low reporting rates also have erroneously low predicted levels of peregrine occurrence.

The authors claim their data mining methods are not sensitive to issues of biased data, but the only basis they give for this claim is citing Kadmon et al. (2004) but Kadmon et al. (2004) does not support their assertion. Kadmon et al. (2004) looks at vegetation in Israel to determine if the fact that much of the data was collected near roads changes the results of bioclimatic models. They conclude that the effect of the bias in data collection is present but small because, in Israel, habitat conditions near roads were similar to habitat conditions away from roads. So, they do not assert that the effect of biased data can be ignored; they examine the underlying data and conclude that the effect of this bias is low in their specific situation. Their bioclimatic models do not however, include distance to road as a variable because that would obviously be affected by roadside bias in the data collection and magnify the effect of that bias. This is however, essentially what this peregrine manuscript does; it uses data with a country-specific reporting bias and then models the results with country-specific predictors.

There is no basis to support the author's claim that these machine learning techniques are impervious to problems of bias in the reported locations. The bias can be minimised with careful model selection, but this paper makes no effort to do so. Without evidence that they overcame the bias in their data, the conclusions are very suspect and this paper should not be published. Perhaps an independent statistical expert can

be consulted on this question.

I also stand by my other criticisms of this manuscript: 1) their testing data does not actually fit their prediction map, the fact that 98% of testing data falls in areas of predicted peregrine use is not informative if 98% of the area is predicted peregrine range; 2) peregrines select for different factors for nesting, wintering, and migration and these cannot and should not be modelled with a single model; and 3) their map does not match independent maps of peregrine distribution (the authors misunderstood this criticism and only stated that Figure 3 and Figure 7 are similar)."

I thank that researcher for taking time to again address both the manuscript and the author responses.

I address these two issues in this comment to ensure an open process even in the most difficult cases. I have taken an editorial decision to reject the manuscript.